# EfficientSkip: Efficiently Transforming Dense LLMs into Sparse Variants

## ABSTRACT

Transformer-based LLMs achieve great success on a variety of NLP tasks, including machine translation, text summarization, and text generation. However, it requires huge amount of computation and data to train such a powerful LLM. Researchers have proposed transformer-based conditional computation algorithms that significantly reduce redundant computations on certain tokens. By skipping dense attention and feed forward computations, these approaches yield sparse LLMs. However, these sparse LLMs are trained from scratch, requiring substantial computation and data. Therefore in this paper, we proposed a training paradigm that can effectively transform a dense transformer-based LLM to its sparse variant with very limited computation resources and merely millions of tokens. We conducted thorough investigations into the key factors that may influence the dense-to-sparse transformation through numerous empirical experiments. In addition, we conducted a case study on the how the tokens skip layers and analyzed their Part-of-Speech tags, gaining valuable insights.

## 1 INTRODUCTION

Transformer-based large language models (LLMs) have been widely used in various application domains due to their great performances. In the near future, LLMs are expected to be deployed on mobile devices, enhancing the lives of more people. However, a vanilla transformer-based LLM requires huge amount of data during the training process and its inference is also computation-intensive and expensive. To this end, researchers investigated a number of transformer variants Kitaev et al. (2020); Zhu et al. (2021); Tay et al. (2022); Pope et al. (2023); Cui et al. (2024) to reduce the training computation and accelerate the inference.

One of the widely used method is conditional computation, which is presented by Bengio (Bengio, 2013). The primary goal of this method is to reduce unnecessary computation. In fact, the hypothesis behind this method is that not all tokens contribute equally to the generation process. This aligns with how humans use language. In many languages, people first consider core words and then add auxiliary words while speaking. The main challenge lies in identifying which token computations are necessary and which are not. To this end, many algorithms are developed to determine when and how much computation should be used in the training and inference (Bengio et al., 2015; Jernite et al., 2016). Instead of processing a token from the input layer all the way to the output layer, an advanced algorithm is used to determine if a token can exit the computation early in the middle layers (Elbayad et al., 2020; Elhoushi et al., 2024). Recently, another algorithm MoD (Raposo et al., 2024) is provided to dynamically allocate the computation across different tokens. In particular, at a certain layer, a router is employed to decide whether a token can bypass the dense attention and feed forward computation.

The MoD-driven model skipped $87.5\%$ of tokens in its optimal configuration while maintaining nearly the same loss as its dense counterpart. However, The MoD models are trained from scratch, which still requires huge amount of computation. Is there a method to obtain a sparse variant of a dense transformer-based LLM without requiring such heavy computation? In this paper, we answer this question by studying a different training paradigm, i.e., transforming a dense LLM into a sparse LLM using very limited data and computational resources, rather than training a sparse LLM from scratch. This paradigm relies on the existing LLM weights, significantly reducing the need for training data and computational resources.

Specifically, we leverage LoRA (Hu et al., 2021) to perform continual pre-training on a small proportion of parameters, transforming an existing dense transformer-based LLM to a relatively sparse LLM with only a few hours of training on merely millions of tokens. Unlike MoD that pre-defines a ratio of how much tokens to skip the layers per sentence, our paradigm skip the layers adaptively so that a pre-defined KL divergence (Hershey & Olsen, 2007) to the base model can be maintained. We also apply binary gates on the hidden states instead of weights, as the latter may cause distortion. The binary gates are specially designed to allow gradients to pass through.

The empirical results show that we can skip the layers of an LLM at a relative low performance cost. In addition, we performed a thorough analysis for the proposed training paradigm, examining the factors that may affect the layer skip rate and model performance. Furthermore, we explored how tokens skip the layers in an example and gained insights from their Part-of-Speech (POS) tags.

To sum up, the main contributions of this paper are as follows:

- We presented a layer skip algorithm that dynamically allocate computation on different tokens.

- We proposed a training paradigm transforming a dense transformer-based LLM to its sparse variant with limited computation on merely millions of tokens.

- We performed a thorough analysis on the proposed paradigm and investigate the factors that influence the dense-to-sparse transformation.

## 2 RELATED WORK

Transformer-based architecture (Vaswani et al., 2017) has been the cornerstone of LLMs. According to Guo (2024), LLM performance scales linearly with the log scale of compute. Hence, training a sophisticated LLM requires a huge amount of FLOPs. This has inspired researchers to explore more efficient transformer architectures (Tay et al., 2022; So et al., 2021). In 2013, Bengio (2013) introduced a concept *conditional computation* which refers to a learned and optimized mechanism that drops out unnecessary compute on some parameters. The concept is explored in depth afterwards (Bengio et al., 2015; Graves, 2016; Jernite et al., 2016).

In recent years, a number of conditional computation methods have been developed for the transformer architecture, that unevenly allocate computation across tokens, thereby saving computation on certain tokens. These methods can be roughly categorized into three groups: 1) "early exit" (Elbayad et al., 2020; Liu et al., 2021; Schuster et al., 2022; Elhoushi et al., 2024), which dynamically ends the computation for tokens and skips the remaining transformer layers, 2) "layer skip" (Raposo et al., 2024), which allows a token to skip continuous or discontinuous middle layers, thereby accelerating forward pass, and 3) "layer branching" (Ainslie et al., 2023), which enables tokens to switch between a light and a heavy branch. Lei et al. (2023) proposed a conditional layer that includes two primary components: a parallel adaptor, as detailed by He et al. (2021), which processes all input tokens, and a pre-trained transformer layer that is specifically designed to process a selected subset of $k$ tokens. The conditional layer enables the conversion of a dense pre-trained model into a sparse model by training on a few parameters.

However, it might require training, either in "pre-training from scratch" mode or "continual pre-training" mode, on billions of tokens to enable the "early exit" mechanism while maintaining the utility of LLMs. The "layer skip" mechanism (Raposo et al., 2024) is investigated solely in the "pre-training from scratch"mode, which is both time-consuming and expensive. "layer branching" requires a well-designed light-weight branch. These inspire us to investigate a mechanism for achieving "layer skip" LLMs with continual pre-training on merely million of tokens via sequential adaptor LoRA (Hu et al., 2021).

## 3 METHODOLOGY

In this section, we describe the model architecture of our approach for skipping layers with routers. We will then demonstrate how to convert a pre-trained transformer to a relatively sparse one with our approach.

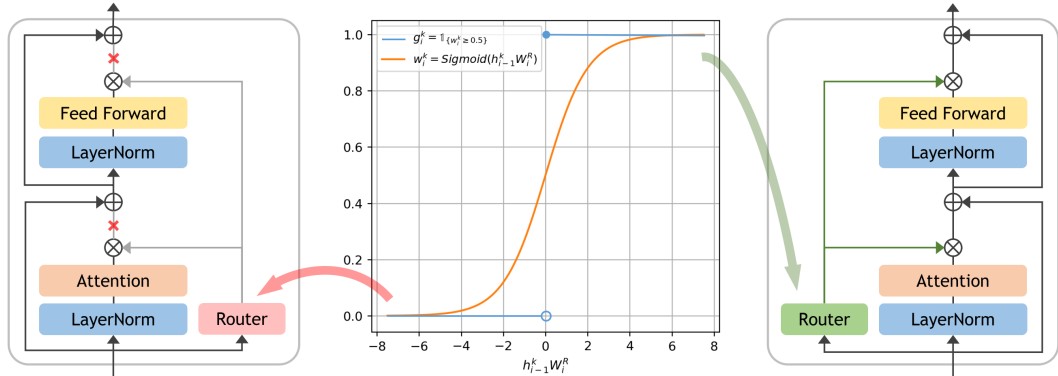

Figure 1: The architecture of one layer from a pre-norm transformer with the router. The normalization occurs before the residual addition. Gates generated by the router are applied to both the attention sub-block and the feed forward sub-block. The left shows the case when the router generate a value of 0. The layer is skipped and the output is the same as the input. The right shows the case when the router generate a value of 1, which functions as a normal transformer layer. The middle shows that the trend of change of $\widetilde{g}_i$ is similar to that of $w_i$. As a result, $g_i$ can borrow the gradients from $w_i$.

### 3.1 MODEL ARCHITECTURE

In transformer, a sequence of tokens are first converted to the corresponding word embeddings $h_0 \in \mathbb{R}^{C \times d}$ by the embedding layer, where $C$ is the number of tokens in the context and $d$ is the hidden dimension. These embeddings are then passed through a stack of $N$ layers $\{L_1, L_2, ..., L_N\}$, where each layer $L_i$ produces its output $h_i$ with the output $h_{i-1}$ from the previous layer as input:

$$h_i = L_i(h_{i-1}) \tag{1}$$

Finally, the last layer hidden states $h_N$ are passed to a task-specific head for down-stream tasks.

Inside each layer $L_i$, there is an attention layer followed by a feed forward layer, each accompanied with a normalization layer and a residual connection. Since our work focus on pre-norm transformers, where the normalization occurs before the residual addition, we can use $A_i$ to represent the sub-block of the attention layer with its preceding normalization and $F_i$ to represent the sub-block of the feed forward layer with its preceding normalization:

$$L_i(h_{i-1}) = h_{i-1} + A_i(h_{i-1}) + F_i(h_{i-1} + A_i(h_{i-1})) \tag{2}$$

To make the transformer relatively sparse, a layer specific router $R_i$ is introduced to decide whether tokens can skip the computation of layer $L_i$. Binary gates $g_i \in \{0, 1\}^C$ are generated for each token in the sequence based on their hidden states $h_{i-1}$ from the previous layer:

$$g_i = R_i(h_{i-1}) \tag{3}$$

With binary gates, the hidden states will not be distorted by the weights.

Unlike in MoD, where the weight generated by the router is applied to the whole layer block output, gates here are applied to each of the individual attention sub-block and the feed forward sub-block as shown in Figure 1. Suppose a value of 0 for $g_i^k$ means the $k$-th token will be skipped for the computation in layer $L_i$. The gated output of the attention sub-block is:

$$g_i \cdot (h_{i-1} + A_i(h_{i-1})) + (1 - g_i) \cdot h_{i-1} \tag{4}$$

which can be simplified to:

$$h_{i-1} + g_i \cdot A_i(h_{i-1}) \tag{5}$$

Similarly, the gated output of the feed forward sub-block, which is also the output of the layer:

$$L_i(h_{i-1}) = h_{i-1} + g_i \cdot A_i(h_{i-1}) + g_i \cdot F_i(h_{i-1} + g_i \cdot A_i(h_{i-1})) \tag{6}$$

## 3.2 ROUTER DESIGN

As mentioned in the previous subsection, the gates generated by the router are binary. We want all gate values to be 1 after initialization, so that no tokens are skipped in any layers, and the model behave exactly the same as the pre-trained transformer. This allows for a functioning initialization for later optimization.

To meet the above requirements, we start with a single linear layer followed by $Sigmoid$ activation:

$$w_i = Sigmoid(h_{i-1}W_i^R) \tag{7}$$

By initializing $W_i^R \in \mathbb{R}^{d \times 1}$ to all 0s, the output weight $w_i$ is a constant vector regardless of $h_{i-1}$. To obtain the binary gate values, each element of $w_i$ is compared against $0.5$:

$$\widetilde{g_i^k} = \begin{cases} 1 & \text{if } w_i^k \geq 0.5 \\ 0 & \text{if } w_i^k < 0.5 \end{cases} \tag{8}$$

and all gate values are initialized to 1.

However, the above implementation cannot propagate gradients back due to the comparison operation. Since the change of $\widetilde{g}_i$ follows the same trend as $w_i$, the gradient of $w_i$ can be borrowed:

$$g_i = \widetilde{g}_i + w_i - sg(w_i) \tag{9}$$

where $sg$ refers to stop gradient. The value of $g_i$ remains binary as the value of $w_i$ and $sg(w_i)$ cancel out. As $\widetilde{g}_i$ and $sg(w_i)$ have no gradients, $g_i$ has the same gradients as $w_i$.

## 3.3 LOSS FUNCTIONS

In order for more tokens to skip a certain layer $L_i$, $g_i$ should have more 0 values. We can use the L1 Norm of $g_i$ as the loss function for this objective:

$$\mathcal{L}_{skip} = \frac{1}{N} \sum_{i=1}^{N} \|g_i\|_1 \tag{10}$$

However, optimization with $\mathcal{L}_{skip}$ alone will cause the model outputs to deviate from the pre-trained transformer outputs. Thus we also introduce the KL divergence loss $\mathcal{L}_{KL}$ to control this deviation.

We found that $\mathcal{L}_{KL}$ tries to reduce the chance for tokens to skip layers, which is opposed to $\mathcal{L}_{skip}$. To balance between the two opposing losses, a KL divergence threshold $t$ is introduced as a hyperparameter. When $\mathcal{L}_{KL}$ reaches or above $t$, $\mathcal{L}_{skip}$ is disabled to prevent from further deviation. It is only enabled when $\mathcal{L}_{KL}$ falls below the threshold:

$$\mathcal{L} = \begin{cases} \mathcal{L}_{KL} + \mathcal{L}_{skip} & \text{if } \mathcal{L}_{KL} < t \\ \mathcal{L}_{KL} + sg(\mathcal{L}_{skip}) & \text{if } \mathcal{L}_{KL} \geq t \end{cases} \tag{11}$$

The $sg(\mathcal{L}_{skip})$ term helps to keep the loss values consistent when alternating between the two cases.

Unlike in MoD, where the skip ratio is predefined, here the skip ratio is a combined effect of the threshold and the input content. The model is trained to skip as many layers as possible as long as it does not deviate above the threshold for the input content. For a certain threshold, the model may have different skip ratio depending on the content. We will see this in Section 4.4.

## 4 EXPERIMENT

In this section, we conduct empirical studies to examine whether the proposed "layer skip" mechanism can transform a dense transformer-based LLM to a more efficient sparse variant with minimal expense. Besides, we also investigate different configurations for such transformation.

## 4.1 DATASET

SlimPajama (Soboleva et al., 2023) is a cleaned and deduplicated version of RedPajama (Computer, 2023). It contains 627B tokens across 59,166 files. For the train split, we create a subset of SlimPajama by selecting 320 files out of these files. We then sample rows from this subset depending on the experiment setup. For the validation split, we sample a fix of 512 rows from a subset of 40 files for all experiment setups.

## 4.2 SETUP

We selected the instruction-tuned version of Gemma 2B (Team et al., 2024) from Google as the subject for continual pre-training in the experiment. In particular, we train for 1 epoch and set the batch size to 4, learning rate to 1e-5 and threshold to 1e-4. The maximum sequence length is set to 288, 576 and 1,152 in different setups.

We apply LoRA on all linear layers of the transformer, including the router. The rank is set to 64, and the trainable parameters take around 3% of the total parameters. The frozen base transformer is quantized to 4-bit as in QLoRA (Dettmers et al., 2023). The pre-training takes about 6 hours for every 1M tokens on one A100 GPU.

## 4.3 EVALUATION METRIC

For evaluation, MT-Bench (Zheng et al., 2023) is used. It consists of 80 high quality questions spanning 8 common categories: writing, roleplay, extraction, reasoning, math, coding, STEM and humanities. We use *gpt-4-0125-preview* as the judge and take the average score of all 8 categories as the performance.

Since we only use KL divergence to guide the pre-training with limited computational data resources, the performance loss is unavoidable. As a result, we propose an evaluation metric termed marginal cost (MC), that measures the cost for per unit gain:

$$MC = \frac{\Delta Perf}{\Delta Skip} \tag{12}$$

where $\Delta Perf$ is the percentage of performance loss relative to the pre-trained transformer with no skip, and $\Delta Skip$ is the percentage of skipped layers out of the total layers.

## 4.4 MAIN RESULTS

We first trained the *gemma-1.1-2b-it* using sampled datasets with varying number of rows and maximum context length to explore whether it is possible to transform a dense transformer-based LLM to a more sparse and efficient variant. The results are shown in figure 2, where the x-axis represents the training data size and the y-axis indicates the MC score.

We observed that it is feasible to perform such transformation with limited data and computation as MC scores can be narrowed down to a low interval given optimal hyper-parameter configurations. For example, the MC scores of the second green point and third yellow point are around 0.2, which means we only lose 20% of a performance unit to gain a full unit of skip. Conversely, poor hyper-parameter configurations can result in low MC scores, making it cost-ineffective to perform such transformation.

We also noted that the 90K training data yielded the lowest MC scores in different context settings. However, merely increasing the size of the training data does not always result in good transformations. The optimal training data size varies across different lines. This does not align well with the empirical scaling law (Kaplan et al., 2020). One possible reason for this is that our continual pre-training aims to approximate the distribution of the base model *gemma-1.1-2b-it*, whereas the pre-training focuses on learning the distribution in the pre-training data. The distribution of the SlimPajama subset may differ from that of learned distribution of the base model. In this case, it is difficult to fit more data given the $\mathcal{L}_{KL}$ restriction.

The results revealed that context length might slightly influence the MC scores. In particular, the average MC scores of the three lines context-288, context-576, and context-1152 are 0.768, 0.804, and 0.794, respectively. The short context-288 achieved the best average MC scores while the middle context-576 achieve the best optimal MC score. The long context-1152 did not further decrease the MC score. The likely reason is that the attention calculation for long samples is more affected by token skipping than for short samples.

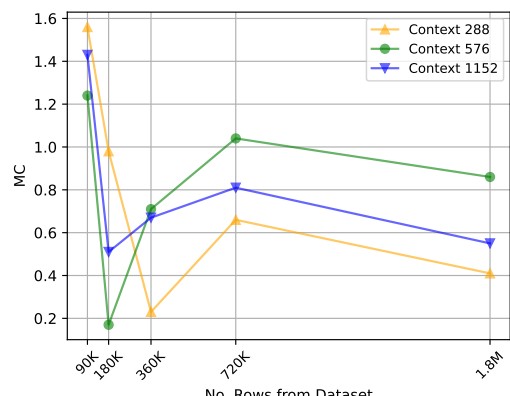

Figure 2: Marginal cost of sampled dataset with 90K, 180K, 360K, 720K and 1.8M rows, and max context length of 288, 576 and 1152.

### 4.5 KL DIVERGENCE AND PERFORMANCE

We use KL divergence to guide the model during continual pre-training. Since a small $\mathcal{L}_{KL}$ indicates that the model is close to the base model, we wonder whether it also performs well on MT-Bench. Figure 3 shows the correlation between $\mathcal{L}_{KL}$ and $\Delta Perf$. Although there are outliers, we can see a positive correlation. We also notice that $\Delta Perf$ becomes more sensitive to $\mathcal{L}_{KL}$ as the latter increases. This allows us to estimate the performance of the model without carrying out the benchmark.

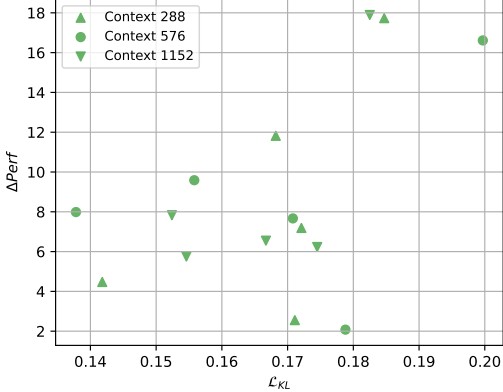

Figure 3: $\Delta Perf$ of experiments with different final $\mathcal{L}_{KL}$. The experiments vary by max context size and dataset size, but they share the same metric of KL divergence to compare with the base model.

### 4.6 EFFECT OF THRESHOLD

The threshold balance between $\mathcal{L}_{KL}$ and $\mathcal{L}_{Skip}$, and we can control $\Delta Skip$ indirectly with it. Figure 4 shows that the change of $\Delta Skip$ with respect to the threshold is close to linear, allowing us to control $\Delta Skip$ effectively with the threshold. MC has the optimal value when the threshold is 1e-4. When the threshold is too small, $\Delta Skip$ is too small to reduce MC. When the threshold is too large, the increase in $\mathcal{L}_{KL}$ cause the model performance to downgrade significantly.

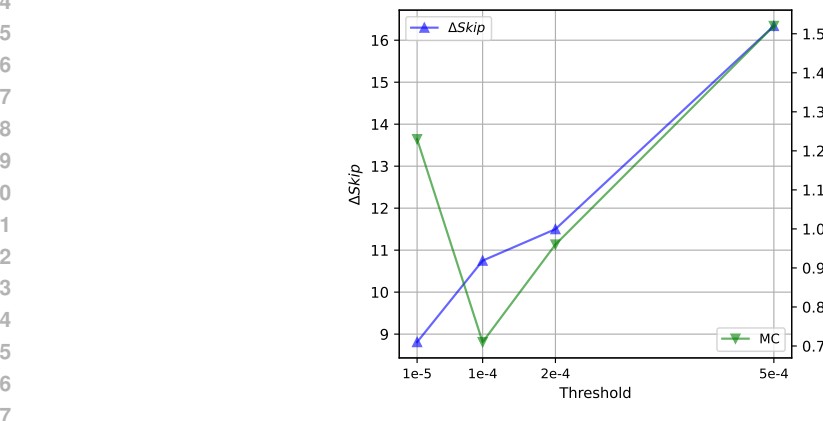

Figure 4: $\Delta Skip$ and MC given the threshold. For all thresholds, a sampled dataset with 360K rows and a max context length of 576 is used.

## 4.7 INFERENCE TIME

We do inference on MT-Bench with different $\Delta Skip$. Figure 5 shows that less time is used compared to the base model as $\Delta Skip$ increases. The amount of reduced inference time is less than $\Delta Skip$ due to overhead. One of the main overhead is the calculation of KV-cache even if the layer is skipped.

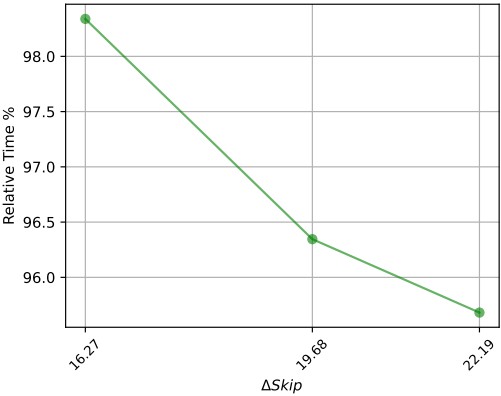

Figure 5: Relative inference time (%) of different $\Delta Skip$ compared to the base model on MT-Bench. Less time is used as $\Delta Skip$ increases.

## 4.8 ABLATION

We conducted several ablation experiments to investigate the effectiveness of the components in our method.

**No Gating on Attention Sub-block**  In our method, $g_i$ is applied on both the attention sub-block and the feed forward sub-block in layer $L_i$. In MoD, the weight generated by the router is only applied on the output of the layer. To see whether the gating on the attention sub-block is necessary, we modified the method as follow:

$$L_i(h_{i-1}) = h_{i-1} + g_i \cdot A_i(h_{i-1}) + g_i \cdot F_i(h_{i-1} + A_i(h_{i-1})) \tag{13}$$

The gates are still binary with borrowed gradients as in our main method. Table 1 shows a 0.71 increase of MC for this ablation, which implies that the gating on the attention sub-block is necessary. A possible reason could be the lack of gradients through the layer for weight optimization.

| Method | MC |
|--------|------|
| Baseline | 0.17 |
| Attention not Gated | 0.88 |
| *K-to-k* Attention | 1.65 |
| Attention Frozen | 0.85 |
| Feed Forward Frozen | 0.90 |

Table 1: MC of different ablations. A sampled dataset with 180K rows and a max context length of 576 is used for all ablations.

**Weights v.s. Binary Gates**  Although weights are widely used in methods like MoD, binary gates are used in our method as weights might scale the hidden states and cause distortion. To verify this claim, we further modify the ablation above as follow:

$$L_i(h_{i-1}) = h_{i-1} + \widetilde{w_i} \cdot A_i(h_{i-1}) + \widetilde{w_i} \cdot F_i(h_{i-1} + A_i(h_{i-1})) \tag{14}$$

and

$$\widetilde{w_i^k} = \begin{cases} w_i^k & \text{if } w_i^k \geq 0.5 \\ 0 & \text{if } w_i^k < 0.5 \end{cases} \tag{15}$$

Experiments showed that the loss even failed to converge during continual pre-training. MoD and most layer skip methods are pre-trained from scratch, and they can learn to adapt to the distorted hidden states during the process. However, our method operates on pre-trained LLMs, and it is hard to tune the pre-trained weights to adapt to the distorted hidden states which they have never seen before.

***K-to-all* v.s. *K-to-k* Attention**  In our method, if a token skips the computation for some certain layers, its hidden state is still used by other tokens to compute attention of that layer. This is termed *k-to-all* attention in CoDA (Lei et al., 2023). On the other hand, MoD does not include the hidden state of a skipped token for attention computation, which is termed *k-to-k* attention. CoDA concludes that *k-to-all* attention has better performance over *k-to-k* attention though it is slower. We experimented with *k-to-k* attention and an MC increase of 1.48 leads to the same conclusion, as the hidden state of the token skipping a layer provides necessary information for the attention of other tokens. The high MC increase suggests that *k-to-all* attention worth the additional computation cost.

**Partial Tune-able Parameters**  The pre-trained weights from the LLM are tuned to adapt to the router during optimization. To see which parts of the parameters play a more essential role in this adaption, we froze the attention sub-block and the feed forward sub-block respectively. The results in Table 1 show a similar increase for both cases. However, we think the attention sub-block is more essential as it has much fewer trainable parameters than the feed forward sub-block. And it plays an active role by adapting to hidden states from non-adjacent layers.

### 4.9 MECHANISM OF THE ROUTER

In pre-norm transformer, the final layer hidden state $h_N^k$ of a token is the sum of the input word embedding $e^k$ and a sequence of vectors $v_i^k$ added by each layer:

$$h_N^k = e^k + \sum_{i=1}^{N} v_i^k \tag{16}$$

where $v_i^k$ represents the following from Equation 2:

$$v_i^k = A_i(h_{i-1}^k) + F_i(h_{i-1}^k + A_i(h_{i-1}^k)) \tag{17}$$

In our method with gates applied, suppose $S$ represent the set of skipped layer indices, the final layer hidden states will be:

$$\widetilde{h_N^k} = h_N^k - \sum_{j \in S} v_j^k \tag{18}$$

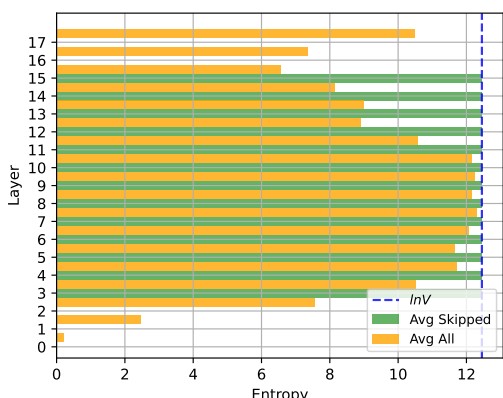

Figure 6: Average per layer entropy of all the tokens generated from a MT-Bench question. Average in green only includes the skipped ones, while average in yellow includes both the skipped and non-skipped ones.

In auto-regressive transformers, the output distribution is obtained from the final layer hidden state with the help of the embedding matrix $E \in \mathbb{R}^{V \times d}$:

$$Softmax(\frac{1}{Z}h_N^k E^T - \frac{1}{Z}\sum_{j \in S} v_j^k E^T) \tag{19}$$

where $V$ is the vocabulary size, and $Z$ is induced by normalization. From the definition of Softmax funcion, if each element in $\sum_{j \in S} v_j^k E^T$ is the same, i.e., $Softmax(\sum_{j \in S} v_j^k E^T)$ is a uniform distribution, skipping layers will not affect the output distribution. We can use entropy to measure whether $Softmax(\sum_{j \in S} v_j^k E^T)$ is closed to uniform. Further more, we wonder whether $Softmax(v_j^k E^T)$ is close to uniform for each $j \in S$ as well.

Figure 6 shows the average per layer entropy of all the generated tokens from a MT-Bench question. The average entropy (orange) corresponding to the skipped ones in each layer is surprisingly close to the maximum $ln(V)$, which implies a uniform distribution from each layer. By contrast, the average entropy (green) of all is close to maximum only in middle layers. This comparison shows that the skipped layers will not affect the output distribution.

We assume after the pre-trained weights are tuned, the router $R_i$ is able to learn to predict whether $Softmax(v_i^k E^T)$ is a uniform distribution from its input $h_{i-1}^k$.

### 4.10 CASE STUDY

We inspect which layers are skipped for the tokens generated from a MT-Bench question and have some interesting findings. The left of Figure 7 shows a small sample from the generation. We find that the skips are most likely to occur on tokens that do not contain much information, e.g., punctuation marks, space and articles. We also find that if a token start skipping a certain layer, it is likely to skip for more than one layer, and in many cases it will skip adjacent layers. Based on the analysis in Section 4.9, the hidden states might contain some information to indicate its non-significance that is recognizable by several routers, causing the token to skip more than one layer.

The right of Figure 7 shows how tokens of each POS tag skip each layer. Numbers are more likely to skip layers than any other POS tags, which indicates for Gemma 2B we are studying, they will not have much influence on the generation. This explains why Gemma 2B is performing badly on math as it is not sensitive to numbers. In contrast, conjunctions are most unlikely to skip layers. The reason is that they affect the trend of the generation, a small error could lead to significant deviation.

Figure 8 shows the per layer percentage of tokens skipping a layer out of all the tokens. We can see the middle layers have the highest percentage, while none of the tokens skip layer 0, 1, 2, 16 and 17. A possible reason could be that the first and last few layers are operating on hidden states that

are more close to the word embedding space, and skipping them can perturb the output distribution significantly. As for middle layers, since they operate on high-level features, they need not to be involved in the generation of every token.

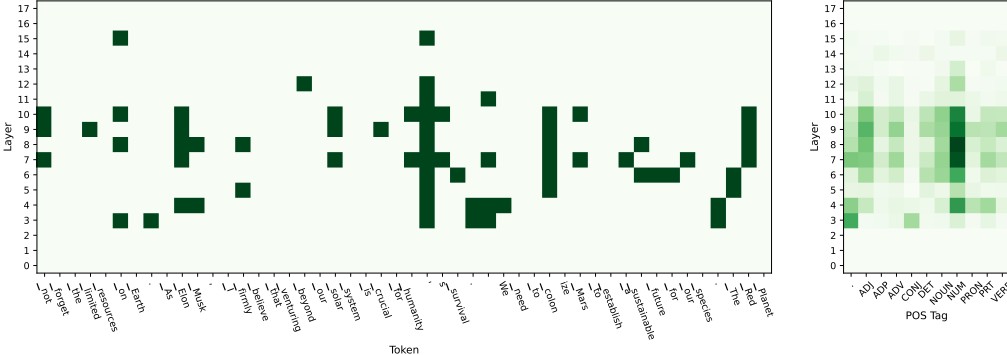

Figure 7: Left: an example of which layers are skipped for each token generated from a MT-Bench question. The dark green squares mark the skipped layer. Right: per layer likelihood of the layer to be skipped for different POS tags. A darker color indicates a larger likelihood.

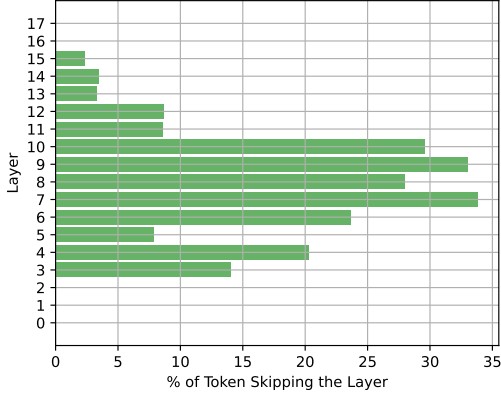

Figure 8: Per layer percentage of tokens skipping a layer out of all the tokens.

## 5 CONCLUSION

In this paper, we propose a training paradigm that effectively transforms a dense transformer-based large language model (LLM) into a relatively sparse LLM. This approach avoids the massive computation and data requirements needed to train a sparse LLM from scratch. Additionally, we utilize the parameter-efficient training method LoRA to significantly reduce trainable parameters and thus accelerate the continual pre-training process. The empirical experiments demonstrate the feasibility and effectiveness of the proposed transformation paradigm. We also investigate different configurations for dense-to-sparse transformation, benefiting the transformation on other LLMs. The case study offers a visual analysis of tokens that are often skipped by routers. Moreover, POS tag analysis of skipped tokens provides deeper insights into the types of tokens that are likely to be skipped.

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
