# OpenReview forum: "EfficientSkip: Efficiently Transforming Dense LLMs into Sparse Variants"
_ICLR.cc/2025/Conference — ICLR 2025 Conference Withdrawn Submission_

### Official Review · Reviewer_jDCC · 2024-11-01

**Soundness:** 2
**Presentation:** 3
**Contribution:** 2
**Rating:** 3
**Confidence:** 5

**Summary:**

Inspired by the sparsity observed in large language models, this paper attempts to convert existing dense models into sparse models. Specifically, it introduces a trainable gate at each layer in the transformer, which controls whether each token in the sequence can skip computation at that layer. Experiments demonstrate that, with training on a small amount of data, their approach can effectively transform a dense model into a sparse model.

**Strengths:**

* The motivation is clear. Numerous studies have established the presence of sparsity in large language models, underscoring the potential of harnessing this sparsity to enhance model efficiency effectively.
* The writing is clear. I can understand this work easily.

**Weaknesses:**

* The novelty is somewhat limited. As is pointed out in this paper, the difference between this work and MoD [1] is the skipping granularity. In detail, MoD skips the total layer while this work skips the sublayer (i.e., attention or feed-forward layer). The improvement is minor. I suggest the authors make a deeper analysis of the reason for this selection.
* The experiments are limited.
  * The model variants and scale are limited. The authors only conducted experiments on Gemma 2B. More open-source popular LLMs such as the LLaMA series and larger sizes such as 7B are necessary to validate the effectiveness of this simple method.
  * The performance evaluation is limited. An important prerequisite for model sparsification is that performance loss should be acceptable. However, the authors provide little coverage on this aspect. More extensive evaluations of the performance are required. For example, they can evaluate their method on popular benchmarks such as MMLU [2], GSM8K [3] and HumanEval [4].


[1] Raposo, David, et al. "Mixture-of-Depths: Dynamically allocating compute in transformer-based language models." arXiv preprint arXiv:2404.02258 (2024).
[2] Hendrycks, Dan, et al. "Measuring massive multitask language understanding." arXiv preprint arXiv:2009.03300 (2020).
[3] Cobbe, Karl, et al. "Training verifiers to solve math word problems." arXiv preprint arXiv:2110.14168 (2021).
[4] Chen, Mark, et al. "Evaluating large language models trained on code." arXiv preprint arXiv:2107.03374 (2021).

**Questions:**

* Why does the paper use a template title for its main title?

---

### Official Review · Reviewer_6t6F · 2024-11-04

**Soundness:** 1
**Presentation:** 1
**Contribution:** 1
**Rating:** 1
**Confidence:** 4

**Summary:**

This paper shows that a dense transformer can be made sparse (using conditional computation proposed in several prior works), via training.

**Strengths:**

The paper provides limited experiments to show their layer skip algorithm can sparsify a dense transformer via training. However, there are several concerns itemized in the weaknesses below.

**Weaknesses:**

1. The title of the paper is incorrect and uses the template title.
2. The motivation and the background of the paper are not clearly laid out at all.
3. Only one model, and a less common choice and size of that, has been evaluated.
4. No explanation for why the SlimPajama dataset was chosen or description of the dataset is given.
5. MT-Bench is the only evaluation dataset used which is insufficient for a comprehensive analysis of model behavior.
6. The notations in Section 4.8 need to be clearly defined for readability.
7. Section 4.10, the case study, is not comprehensively analyzed and the conclusions drawn are shallow.
8. Sentences, such as the one in lines 247-249, are overly long and not punctuated.
9. Several typos, such as line 208: "for such *a transformation"

**Questions:**

Why was only one model and only one training and evaluation task used for the experiment?

---

### Official Review · Reviewer_PYcZ · 2024-11-04

**Soundness:** 2
**Presentation:** 3
**Contribution:** 3
**Rating:** 3
**Confidence:** 4

**Summary:**

This paper comes up with a method of converting a dense pretrained LLM into a sparse LLM in an efficient manner without retraining from scratch and using just mere millions of tokens. This is achieved by adding routers to each layer which dynamically decide whether a token should skip a particular layer or not. Then they train the model further and use KL divergence in a clever way to ensure that the model does not deviate from it's initial pretrained outputs.

**Strengths:**

1. The method suggested here seems to be pretty interesting including the choice to use L1 loss and KL loss.

2. I also really enjoyed reading sections 4.8, 4.9 and 4.10

**Weaknesses:**

1. I believe the paper lacks the required volume of experiments needed for a conference like ICLR. I would have loved if the authors could evaluate on another benchmark like say GSM-8k or MMLU-Pro.

2. I also believe the choice of using Gemma 2B with context lengths of 288, 576 and 1152 is pretty odd. I realize the there are compute constraints in academia but it would have really been great if there were results on an 7-8B model since 2B models are hardly used in my experience. In general I would have liked if the paper demonstrated the efficacy of the method on atleast one more model and one more evaluation dataset.

3. The finding that the MC drops when we switch from 288 to 1152 context is a bit concerning to me since I believe long context is the future and if a method cannot handle long context very well it's a bit concerning. Also I think sparse LLMs would be much more useful when the computation costs are higher and computation costs are generally higher with longer examples. So I would love for the authors to dive a bit deeper into what exactly is happening at longer contexts. Maybe look individually at $$\Delta skips$$ and $$\Delta performance$$ and try to investigate a bit further. I am not really satisfied by the one line explanation the current paper has.

4. I think a comparison with the baseline of some layer pruning techniques would have been great. I believe this method should work better than skipping a whole layer entirely for all tokens but still a baseline comparison would have been great.

5. A small nitpick. I would have loved if the captions were a bit more informative.

**Questions:**

Please see weaknesses

1. I would also like to see how much faster is the best performing model than the original model and how much performance it loses. As much as MC makes sense, it would be great to see a table where we can clearly see the amount of time saved and delta in performance separately for the best model.

---

### Official Review · Reviewer_5bez · 2024-11-07

**Soundness:** 2
**Presentation:** 1
**Contribution:** 2
**Rating:** 3
**Confidence:** 4

**Summary:**

This paper proposes a method to transform a dense transfer to a sparser version using LoRA based continued pre-training. The method introduces binary gates on hidden states instead of the weights to selectively skip computation at specific layers. The authors propose the use of a KL-divergence based loss function to prevent the weights deviating too much from the pre-trained weights. The authors perform experiments on the Gemma 2B Instruct model using a subset of the SlimPajama dataset.

**Strengths:**

- The proposed method addresses an important research problem of computational cost associated with training sparse LLMs. More often than not, sparse LLMs have to be trained from scratch.
- The proposed approach supports gating within a layer on the attention or the feedforward sub block supporting more granular sparsity instead of skipping an entire layer.

**Weaknesses:**

- There are gaps in writing making understanding a bit hard, even the paper title is wrong in the pdf. I would suggest the authors to proofread properly and correct the various typos in citations, for example line 35: which is present by Bengio (2013) rather than presented by Bengio (Bengio, 2013). I would suggest the authors to understand the differences between \citet and \citep for accurately citing the references.
- The experiments are quite limited: a single small model is used, the sequence length is very small to understand the nuances, only a subset of the pre-training dataset is used, and finally only one benchmark - MTBench is used for the analysis. MTBench in itself is quite flawed because of the limited number of samples in the benchmark.
- No baselines are present in the paper to compare other sparse LLMs.

**Questions:**

I would suggest the authors to significantly improve the paper to be considered an ICLR-level submission. But it's a great start for a research project and relevant for some workshop paper. I would suggest the authors to incorporate the feedback in the weaknesses to improve their paper.

---

### Note · Authors · 2024-11-26

I have read and agree with the venue's withdrawal policy on behalf of myself and my co-authors.